## [Decision Letter · Decision Letter 0]

7 Oct 2025

Histone demethylase JMJD1A protects mice from enteric bacterial infection by upregulating CCL8 expression to recruit macrophages and CD4+ T cells

PLOS Pathogens

Dear Dr. YU,

Thank you for submitting your manuscript to PLOS Pathogens. Your manuscript was evaluated by members of the editorial board and three external referees. All were enthusiastic about your study, but there were some major issues with the writing and experimentation. Therefore, we invite you to submit a substantially revised version of the manuscript that addresses the all of the points raised by all three reviewers; in particular all were concerned about validation of the the role of CCL8: Reviewers 2 and 3 have detailed experiments regarding restoration of CCL8 that is required for a revised manuscript.

Please submit your revised manuscript within 60 days of 10/06/2025. The revision would be due by 12/6/2025. If you will need more time than this to complete your revisions, please reply to this message or contact the journal office at plospathogens@plos.org. Please include the following items when submitting your revised manuscript:

We look forward to receiving your revised manuscript.

Kind regards,

George Deepe, jr., MD

Academic Editor

PLOS Pathogens

D. Scott Samuels

Section Editor

PLOS Pathogens

Sumita Bhaduri-McIntosh

Editor-in-Chief

PLOS Pathogens

orcid.org/0000-0003-2946-9497

Editor-in-Chief

PLOS Pathogens

orcid.org/0000-0002-7699-2064

**Journal Requirements:**

Potential Copyright Issues:

i) Figure 7I. Please confirm whether you drew the images / clip-art within the figure panels by hand. If you did not draw the images, please provide (a) a link to the source of the images or icons and their license / terms of use; or (b) written permission from the copyright holder to publish the images or icons under our CC BY 4.0 license. Alternatively, you may replace the images with open source alternatives. See these open source resources you may use to replace images / clip-art:

5) Thank you for stating that "The sequencing raw data in the present study have been deposited in the National Center for Biotechnology Information (NCBI) database, and the project numbers are PRJNA1093258 (Our SRA records will be accessible with the following link after the indicated release date: https://www.ncbi.nlm.nih.gov/sra/PRJNA1093258) and PRJNA1094454 (Our SRA records will be accessible with the following link after the indicated release date: https://www.ncbi.nlm.nih.gov/sra/PRJNA1094454)." Please note that, though access restrictions are acceptable now, your entire minimal dataset will need to be made freely accessible if your manuscript is accepted for publication. This policy applies to all data except where public deposition would breach compliance with the protocol approved by your research ethics board.

6) In the online submission form, you indicated that "the data that support the findings of our study also can be obtained from the corresponding author (cdyu@xmu.edu.cn)." All PLOS journals now require all data underlying the findings described in their manuscript to be freely available to other researchers, either

1. In a public repository

2. Within the manuscript itself

3. Uploaded as supplementary information.

7) Please amend your detailed Financial Disclosure statement. This is published with the article. It must therefore be completed in full sentences and contain the exact wording you wish to be published.

8) Please amend your 'Competing Interests' statement in the online submission form and declare all competing interests beginning with the statement "I have read the journal's policy and the authors of this manuscript have the following competing interests:"

**Reviewers' Comments:**

Reviewer's Responses to Questions

**Part I - Summary**

Reviewer #1: This manuscript investigates the role of the histone demethylase JMJD1A in host defense against Citrobacter rodentium infection. The authors demonstrate that JMJD1A promotes host survival by regulating the expression of IRF1 and CCL8, which, in turn, enhances the recruitment of macrophages and CD4+ T cells, key for pathogen clearance. The study is mechanistically sound, relevant, and provides novel insight into epigenetic regulation and host-pathogen interactions. However, some aspects require clarification and expansion before the manuscript can be considered for publication.

Reviewer #2: In this manuscript, Lin et al. identify JMJD1A as a key epigenetic regulator of host defense against Citrobacter rodentium infection. The authors demonstrate a striking phenotype in which JMJD1A-deficient mice uniformly succumb to infection, exhibiting severe colonic pathology, impaired bacterial clearance, and systemic dissemination. Mechanistically, JMJD1A expression is induced via IRF1 during infection, where it cooperates with STAT1 and IRF1 to promote CCL8 expression, thereby enhancing macrophage and CD4+ T cell recruitment and supporting protective antibody responses. This represents a strong and rational approach that links chromatin regulation to mucosal immune defense through a defined molecular pathway. The data are convincing, and the experimental approaches are well designed and executed. While the findings are compelling, additional experiments such as deeper validation of epithelial-intrinsic effects of JMJD1A in AAV9 mouse models, combined loss and gain of function analyses to dissect the direct relationship between JMJD1A and CCL8 during infection, and a broader discussion of epigenetic modifiers in regulating enteric infections would further strengthen the mechanistic impact and expand the study’s significance.

Reviewer #3: The manuscript entitled “Histone demethylase JMJD1A protects mice from enteric bacterial infection by upregulating CCL8 expression to recruit macrophages and CD4+ T cells” by Lin et al. reports a novel role for JMJD1A in host defence against Citrobacter rodentium infection. The authors demonstrate that JMJD1A enhances the expression of IRF1 and CCL8, which in turn promote the recruitment of macrophages and CD4+ T cells to the colon following infection, facilitating bacterial clearance. Mice lacking JMJD1A exhibit elevated colonic and systemic bacterial burdens and ultimately succumb to infection. The study combines in vitro analyses using the CT26 cell line with in vivo approaches, including AAV9-mediated delivery of shRNA and protein DNA constructs to colonic epithelial cells, to dissect the roles of JMJD1A, CCL8, and IRF1 during infection.

Overall, the manuscript provides strong experimental evidence to support the conclusions. I have the following comments that may help further strengthen the quality and impact of the work:

**Part II – Major Issues: Key Experiments Required for Acceptance**

Reviewer #1: 1. The manuscript focuses on epithelial JMJD1A, but knockdown and overexpression via AAV do not exclude roles in immune cells. A conditional knockout or a focused discussion of this limitation is needed.

2. Although the data strongly implicates CCL8, the conclusion that JMJD1A's protective effect relies entirely on CCL8 may be an overstatement. The authors should explore other pathways that operate downstream of JMJD1A.

3. The observed reduction in IgG and IgA suggests potential defects in adaptive immunity. However, it is unclear whether this is solely due to a reduction in CD4+ T cells or reflects intrinsic effects on B cells. The discussion should address this uncertainty more directly.

4. The results from 16S rRNA sequencing are presented briefly and remain correlational. There is a lack of functional validation, such as through fecal transplants or antibiotic treatments. At a minimum, the discussion should elaborate on whether the differences in the microbiome are a cause or a consequence of infection susceptibility.hould expand on whether microbiome differences are a cause or consequence of infection susceptibility.

Reviewer #2: 1) Validation of AAV9-knockdown and over-expression in only intestinal epithelial cells, but not tissue-associated immune cell populations. Could the author include IHC staining for JMJD1A and CCL8 in tissue, and quantify loss/gain of expression in only epithelial cells, but similar levels in underlaying lamina propria populations? In order to claim that these are epithelial-specific expression, confirmation is required.

2) The authors demonstrate that JMJD1A knockout mice have reduced CCL8 levels and increased Citrobacter burdens and infection-induced pathology, and that overexpression of CCL8 reduces bacterial burdens and histopathology. They therefore conclude that JMJD1A protects against infection through CCL8. To fully support this claim, could the authors restore CCL8 expression using AAV-mCCL8 constructs or exogenously supplement CCL8 in JMJD1A knockout mice to directly demonstrate that CCL8 protects these mice from infection? This is a key conclusion of the paper. While the authors clearly show using in vitro molecular approaches that JMJD1A regulates CCL8 transcription, they have not tested whether CCL8 is sufficient to protect JMJD1A-deficient mice from infection.

3) In addition to JMJD1A, several histone-modifying enzymes such as LSD1, JMJD2D, EZH2, and HDAC3 have been investigated for their roles in regulating intestinal immunity and protection against enteric infection. A more thorough discussion of these studies would help place the current findings in better context within the field and highlight how this work advances our understanding of epigenetic regulation in host defense.

Reviewer #3: Major Comment 1:

AAV9-mCCL8 experiment in JMJD1A-/- mice: The authors performed AAV9-mediated expression of CCL8 in colonic epithelial cells of WT mice following C. rodentium infection (Fig. 5E). Overexpression of CCL8 reduced disease severity, as evidenced by improved colonic histology and decreased colon-associated CFU. In addition, CCL8 overexpression promoted the recruitment of macrophages and CD4+ T cells to the colon and increased C. rodentium–specific IgG and IgA levels in the faeces. These findings mirror, in the opposite direction, the phenotypes observed in JMJD1A-/- mice.

While this experiment highlights the importance of CCL8 in driving immune cell recruitment during infection, its relevance to the disease observed in JMJD1A-/- mice remains indirect. To firmly establish that the severe disease in JMJD1A-/- mice results from impaired CCL8 expression, it would be essential to test whether AAV9-mediated CCL8 expression in JMJD1A⁻/⁻ mice can rescue the infection outcomes. Such an experiment would directly corroborate the proposed mechanistic link between JMJD1A, CCL8, and host protection.

Major comment 2:

Analysis of reduced IFNγ levels in JMJD1A-/- mice: As shown in Fig. 5A, JMJD1A-/- mice exhibit lower colonic IFNγ levels. Since IFNγ is a well-established upstream driver of STAT1 phosphorylation and subsequent IRF1 activation (DOI: 10.1038/s41467-022-33326-5), it would be important to determine whether reduced IFNγ contributes to the impaired STAT1 and IRF1 axis in JMJD1A-/- mice. I recommend that the authors should quantify IFNγ levels and assess STAT1 phosphorylation in JMJD1A-/- mice, similar to what is shown for WT mice in Fig. 6F. Such data would clarify whether diminished IFNγ is an independent factor underlying reduced STAT1 and IRF1 activity in the absence of JMJD1A.

Major comment 3:

Data presentation and statistical analysis – The figure legends state that the data are representative of three independent experiments. It would strengthen the study to pool data from all replicates and present individual data points, allowing readers to better appreciate the inherent biological variability. In addition, Student’s t-test is applied throughout, including in cases where more than two groups are compared. In such cases, an appropriate statistical test with correction for multiple comparisons should be used to avoid potential bias.

**Part III – Minor Issues: Editorial and Data Presentation Modifications**

Reviewer #1: 1. Please add discussion points on the relevance of this study to human infections, such as implications for EPEC/EHEC infection and translational potential of targeting JMJD1A.

2. Address whether AAV-mediated modulation itself could alter immune responses- what are the implications of AAV’s off-target effects on the microbiome?

3. Revise the description of AMPs from AMP-related genes to AMPs

4. Line # 235: revise to JMJD1A-mediated protection against C. rodentium is not through regulation of AMP gene expression.

5. Line # 450: revise to CCL8-deficient mice were more susceptible to Listeria monocytogenes infection.

6. Line # 453: revise to after infection

Reviewer #2: 1) The authors should be careful not to overinterpret their results. They convincingly show that JMJD1A deficiency and CCL8 overexpression alter macrophage and CD4 T cell numbers as well as Citrobacter-specific antibody responses. However, they do not include functional loss- or gain-of-function experiments to directly demonstrate that increasing these cell populations is sufficient to confer protection. Moreover, since JMJD1A is also lost in immune cells in the knockout model, it remains unclear whether changes in cell numbers alone explain the phenotype. I encourage the authors to carefully review the manuscript to ensure that the conclusions remain aligned with the data presented. It is also possible that additional JMJD1A-dependent protective pathways contribute to host defense but have not yet been tested.

2) The authors include microbiome analyses of JMJD1A WT and KO mice in the supplemental data, but these results are introduced only in the discussion section. To improve clarity, these data should be referenced in the results section and then further contextualized in the discussion. Presenting the findings in this way would provide a more logical flow and strengthen the overall narrative of the manuscript.

3) To improve clarity, could the authors increase the size of the IHC images and provide higher-magnification views? This would allow better assessment of staining quality and more accurate evaluation of cell numbers within the tissue.

Reviewer #3: Minor Comments

1. Quantification of Western blots: Since the western blots shown in the figures are representative of three repeats, it would strengthen the data to include densitometric analysis for Fig. 1A, 1G, 1J, and 3H.

2. Heat-killed CR and JMJD1A elevation: Heat-killed C. rodentium induces the elevation of JMJD1A, IRF1, and CCL8 in CT26 cells. The authors should discuss what might activate these pathways, as this response is clearly independent of the A/E feature of the bacterium.

3. Fig. 1H: An Excel sheet with all transcription factors should be provided. The use of “…..” should be avoided in Fig. 1H.

4. IHC: When commenting on the deep crypt localization of C. rodentium in different models, the authors should provide a zoomed-in image of the crypts for better visualization and appreciation of the phenotype. The current IHC images in Fig. 3G, 3K, and 5H do not confidently support the stated conclusions.

5. H&E staining in Fig. 3K: The representative H&E images shown for day 14 post-infection depict two mice with similar damage and CCH, which contradicts the scoring and the conclusion.

6. Survival and weight loss: Can the authors add survival and weight loss data for the AAV9-shJMJD1A model? This would allow a direct comparison between the whole-body knockout and the colonic epithelial knockdown using the AAV9 system.

7. Missing figure legends: Legends for Fig. 4A–C are missing.

8. Microbiota data: The microbiota data in Supplementary Fig. 5 are not described in the Results section but are discussed in the Discussion. They should be included in the Results.

9. CD4⁺ T cell cytokines: Can the authors measure cytokines produced by CD4⁺ T cells?

10. Lines 487–489: The explanation provided does not adequately address the contradictory results. IRF1 KO mice develop severe disease upon C. rodentium infection, while conditional epithelial-specific IRF1 KO mice display comparable disease. In the AAV9-shIRF1 model, mice also exhibited severe disease. However, the statement that constitutive epithelial IRF1 KO may involve compensatory mechanisms does not explain why these mice would still develop severe disease.

PLOS authors have the option to publish the peer review history of their article (what does this mean? ). If published, this will include your full peer review and any attached files.

**Do you want your identity to be public for this peer review?** For information about this choice, including consent withdrawal, please see our Privacy Policy .

Reviewer #1: No

Reviewer #2: No

Reviewer #3: No

**Figure resubmission:**

**Reproducibility:**



---

## [Editor Report · Decision Letter 1]

18 Feb 2026

Dear Dr. Yu,

We are pleased to inform you that your manuscript 'Histone demethylase JMJD1A protects mice from enteric bacterial infection by upregulating CCL8 expression to recruit macrophages and CD4+ T cells' has been provisionally accepted for publication in PLOS Pathogens.

Best regards,

George Deepe, jr., MD

Academic Editor

PLOS Pathogens

D. Scott Samuels

Section Editor

PLOS Pathogens

Sumita Bhaduri-McIntosh

Editor-in-Chief

PLOS Pathogens

orcid.org/0000-0003-2946-9497

Michael Malim

Editor-in-Chief

PLOS Pathogens

orcid.org/0000-0002-7699-2064

---

## [Editor Report · Acceptance letter]

Dear Dr. Yu,

We are delighted to inform you that your manuscript, "Histone demethylase JMJD1A protects mice from enteric bacterial infection by upregulating CCL8 expression to recruit macrophages and CD4+ T cells," has been formally accepted for publication in PLOS Pathogens.

Best regards,

Sumita Bhaduri-McIntosh

Editor-in-Chief

PLOS Pathogens

orcid.org/0000-0003-2946-9497

Michael Malim

Editor-in-Chief

PLOS Pathogens

orcid.org/0000-0002-7699-2064